# Improved High-Yield PMMA/Graphene Pressure Sensor and Sealed Gas Effect Analysis

**DOI:** 10.3390/mi11090786

**Published:** 2020-08-19

**Authors:** Ying Liu, Yong Zhang, Xin Lin, Ke-hong Lv, Peng Yang, Jing Qiu, Guan-jun Liu

**Affiliations:** 1College of Intelligence Science and Technology, National University of Defense Technology, Changsha 410073, China; liuying@nudt.edu.cn (Y.L.); zhangyong@nudt.edu.cn (Y.Z.); linxin162@163.com (X.L.); fhrlkh@163.com (K.-h.L.); nudtyp_7984@163.com (P.Y.); qiujing@nudt.edu.cn (J.Q.); 2Science and Technology on Integrated Logistics Support Laboratory, Changsha 410073, China

**Keywords:** pressure sensor, graphene, PMMA, sealed gas

## Abstract

Graphene with atomic thickness possesses excellent mechanical and electrical properties, which hold great potential for high performance pressure sensing. The exposed electron of graphene is always cross-sensitive to any pollution absorbed or desorbed on the surface, from which the long-term stability of the graphene pressure sensor suffers a lot. This is one of the main obstacles towards graphene commercial applications. In this paper, we utilized polymethylmethacrylate (PMMA)/graphene heterostructure to isolate graphene from the ambient environment and enhance its strength simultaneously. PMMA/graphene pressure sensors, with the finite-depth cavities and the through-hole cavities separately, were made for comparative study. The through-hole device obtained a comparable sensitivity per unit area to the state of the art of the bare graphene pressure sensor, since there were no leaking cracks or defects. Both the sensitivity and stability of the through-hole sensor are better than those of the sensor with 285-nm-deep cavities, which is due to the sealed gas effect in the pressure cavity. A modified piezoresistive model was derived by considering the pressure change of the sealed gas in the pressure cavity. The calculated result of the new model is consistent with the experimental results. Our findings point out a promising route for performance optimization of graphene pressure sensors.

## 1. Introduction

Graphene, as natural atomic material, is an ideal option as a sensing membrane of pressure sensors. Pressure sensors are used in a tremendous range of applications, from our daily life to industrial monitoring [1,2]. The core component for the pressure sensor is the sensing membrane, which transduces the applied pressure to an electrical signal. Most traditional pressure sensors employ thin bulk materials [3] as membrane, like either silicon-based [4] or polymer-based [5] sensing membranes. These kinds of membranes are usually thinned by chemical etching or directly deposited, which results in a rough surface. Another problematic issue is the thickness of bulk membranes on the order of microns. To obtain sufficient pressure sensitivity, therefore, traditional pressure sensors need large-size membranes of several hundreds of microns. Reducing the thickness of the sensing membrane becomes the major path towards higher performance pressure sensors. Graphene comes to the center of the stage because of its atomic thickness reaching the limit of physics feasibility [6,7]. Meanwhile, graphene possesses extraordinary mechanical and electrical properties. Its Young’s modulus is up to 1 TPa [8], which makes it the strongest material today. It can be stretched to 20% [8] and is flexible for large deflection. Inspiringly, the band structure of graphene can be tuned by strain [9,10], i.e., piezoresistive effect, which makes the graphene both the membrane for pressure loading and the transduction for direct electrical readout simultaneously. All these merits above enable graphene to have a promising future for high performance pressure sensors.

The graphene-based piezoresistive pressure sensor has been demonstrated by several research groups worldwide in recent years. The first prototype was built by J. Scott Bunch et al. [11] to test the impermeability of the graphene membrane by applying pressure difference across the membrane. A. D. Smith et al. [12] later made the first pressure sensor based on suspended graphene membranes with direct electrical readout utilizing the piezoresistive effect in graphene. It has demonstrated that the sensitivity per unit area of the graphene sensor is about 20 to 100 times higher than that of conventional piezoresistive pressure sensors, which is encouraging to both the academic community and the industrial world. Even though there are lots of similar works that followed [13,14,15], in reality, the graphene pressure sensor suffers a lot from issues of low yield, membrane cracks, and gas leakage, since the fabrication processes bring a heavy load on the atomic graphene sheet. To overcome these problems, we introduced a polymer-assisted graphene pressure sensor scheme where a thin poly-methylmethacrylate (PMMA) film was used as both a supporting and sealing layer of the graphene sheet [16].

In this work, we applied our polymer-assisted scheme to two different pressure sensor structures to study the sealed gas effect in the pressure cavity. One is the most reported structure, in which the pressure cavity has a finite depth. The other is with a through-hole design. Their performances were measured and analyzed. An improved piezoresistive model was built to explain the increased performance in the sensor with the through-hole cavity.

## 2. Device Fabrication and Measurement Setup

### 2.1. Device Fabrication

To study the effect of the sealed gas in the pressure cavity, PMMA/graphene pressure sensors with a 285-nm-deep cavities and through-hole cavities were designed on the same chip. The fabrication process flow is schematically depicted in Figure 1.

First, the silicon substrate, capped with thermally grown 285 nm-thick SiO2, was cleaned and pretreated with Hexamethyldisilazane (HMDS) to modify its surface hydrophilicity, which was followed by spin-coating a layer of AZ1500 photoresist. Standard ultraviolet lithography was applied to transfer the pressure cavity pattern to the photoresist. The cavities of 285 nm depth were etched by buffered oxide etchant (BOE). To make the through-hole, we introduced twice reactive ion etch (RIE) processes, since the substrate was too thick to etch through. Firstly, a new layer of photoresist was spin-coated after removing the old resist. Then, only the through-hole cavities were patterned. After the first RIE process, about 40-μm-thick Si in the cavity was removed. Then, it was turned upside down to apply the second RIE process. A thicker photoresist with 7 μm thickness was used as a mask for a long time RIE. Rectangle windows with size larger than the through-hole cavity regime were patterned. After the second RIE process, all cavities connected to the backside windows formed through-hole structure, as shown in Figure 1e. Next, the electrical leads, consisting of 5 nm Cr and 200 nm Au, were patterned by standard lithography and deposited by thermal evaporation techniques. At this point, the substrate was prepared ready for the graphene transfer.

Since graphene is only one atom layer, protecting graphene from cracks and collapse is one of the most important issues towards high yield fabrication and practical applications. Herein, instead of removing the PMMA support layer after transfer, as described in the traditional transfer process in most articles, we kept the thin layer of PMMA forming PMMA/graphene heterostructure to enhance the suspended membrane strength. The transfer flow of PMMA/graphene heterostructure is presented in Figure 2. As shown in Figure 2a, the CVD graphene on copper foil was fixed on a glass wafer with tape along the edges. By controlling the spinning speed, a thin PMMA layer of about 50 nm-thick was covered on graphene and baked at 120 °C for 90 s. This PMMA can not only provide graphene enough strength, but also isolate it from the harmful ambient environment that would deteriorate its quality. The Cu/graphene/PMMA was floating on Cu etchant (1 M FeCl_3_) to remove the copper, followed by repeatedly rinsing in deionized water (DI). Then, the cleaned PMMA/graphene (PMMA/Gra) membrane was scooped up by the prepared substrate and baked at 150 °C for several minutes to remove the interface water between graphene and the substrate, and also make the membrane stick to the substrate surface well. The transferred membrane is shown in Figure 2f.

To pattern graphene, we introduced a bilayer resist as the mask, which consists of a LOR 10A resist layer and a AZ 1500 resist layer. Experimental tests found that the LOR-10A resist is well compatible with PMMA, while AZ 1500 will attack the PMMA layer. Thus, a layer of LOR-10A resist was inserted between AZ 1500 and PMMA for isolation. Developer 3083 was well chosen as both a developer and remover for the bilayer mask because it is a good remover for LOR. The graphene pattern was transferred by ultraviolet lithography. After O_2_ plasma etching, the LOR and AZ 1500 bilayer was removed by rinsing in 3038 solvent. Finally, the sample was cleaned in DI water and blown dry gently using N_2_ gas. Figure 3a shows the optical image of the fabricated device. In Figure 3b, the device was bonded on a commercial pressure sensor stage for testing.

All the processes above are compatible to the traditional CMOS process and suitable for large-scale fabrication, which holds great potential to bridge lab research to commercial application. Following our fabrication scheme, our PMMA/graphene pressure sensor fabrication achieves an almost 100% yield.

### 2.2. Measurement Setup

Our measurement scheme is illustrated in Figure 4. We introduce a commercial pressure sensor as a reference. Each sensor had a response to the pressure change in our homemade pressure control system (PCS). The data were recorded for performance comparison. The measurement scheme and practical setup can be seen in Figure 4. The homemade PCS had four testing ports for the sensor installation and the gas channel was linked to a clean nitrogen source. The PCS was repeatedly refreshed with nitrogen gas before every test to avoid disturbances from ambient adsorbates. The pressure inside can be manually tuned by rotating a lever arm from 0 to 80 kPa. One commercial pressure sensor (PC10 series, Yuzhidu Co., Ltd., Nanjin, China) was set as a reference sensor and recorded the real-time pressure, which is labelled as 2#. Our packaged sensor labeled as 1# was installed close to 2#. Since the response signal from the commercial 2# has already been well modulated, the data were directly recorded by a multimeter. Meanwhile, our PMMA/graphene sensor without a conditioning circuit generated a weaker signal, which was easily overwhelmed by thermal and electromagnetic noises. We introduced the Wheatstone bridge resistance measurement scheme to reduce the electromagnetic noise, and the bridge was biased with square wave pulses to reduce the current thermal effect. The unbalanced output voltage of the Wheatstone bridge was the response of the sensor to the applied pressure difference and was measured by another multimeter. All the data were sent to the computer through the remote communication mode in real-time.

## 3. Results and Discussion

### 3.1. Improved Performance of PMMA/Graphene Pressure with Through-Hole

During several venting and pumping cycles, the time-dependent responses of both sensors were monitored, as shown in Figure 5. The red curve is the chamber pressure in PCS extracted from the reference commercial sensor. The dark curve is the resistance of the PMMA/graphene pressure sensor with through-holes. A preliminary impression drawn from Figure 5 is that the resistance response of our graphene pressure sensor follows the pressure change very well, especially at these valleys and peaks, which means that our sensor operates at a comparable response speed as the commercial one.

For quantitative assessment, we kept the pressure constant at several steps and measured the time trace of each sensor (Figure 6a). Under each pressure step, the averaged resistance and the corresponding standard deviation were extracted. As shown in Figure 6b, we obtain the nonlinear relationship of the graphene resistance and applied pressure similar to most literature reports [12,17]. Previously, this nonlinearity was attributed to the gas leakage. In our case, it can be drawn that the nonlinear characteristic may be the nature of the pressure-strain model rather than the gas leakage from graphene surface defects, since the graphene is well covered by the PMMA resist. The measured resistance exhibits a fluctuation with a maximum standard deviation of 0.25 Ω, as shown in Figure 6b. One of the possible deviation sources may be the electromagnetic noise coupled to our measurement system. This may be eliminated by an integrated design of the sensor and measurement circuit on one chip in the near future. Further, the sensitivity is estimated according to the typical model for piezoresistive membrane-based pressure sensors, as given by Equation (1):(1)$$S=\frac{∆R/R}{∆P},$$
where *S* is the sensitivity, ∆P the change of pressure, ∆R the corresponding resistance change, and *R* the initial resistance. The sensitivity of the PMMA/Gra pressure sensor is calculated to be *S* = 7.42 × 10^5^ kPa^−1^. Table 1 lists the major published results of several typical piezoresistive membrane-based pressure sensors. For comparison, the sensitivity was normalized to the membrane area in the last row of the table. Graphene-based pressure sensors are at least one order better than conventional silicon-based and CNT-based pressure sensors. Our PMMA/Gra pressure sensor is comparable and a little more sensitive than that of the M.C. Lemme group [12], even though the PMMA/Gra membrane is much thicker than the bare graphene membrane. A reasonable explanation could be the membrane integrity. The graphene, as the atomic membrane, is very fragile and easily becomes cracked after violent fabrication processes [18,19]. For CVD graphene, the unavoidable pinholes or boundary defects from synthesis are another natural source breaking the membrane integrity [20,21]. The atmosphere isolated by the graphene membrane is partially connected, which results in the deterioration of the sensitivity per unit area. In our device, the defects of the graphene sheet layer are fully sealed by the PMMA layer and no channel for leakage. The improved performance due to the absence of leaks implies that the leakage of graphene membrane may have a much worst effect than what has been reported in previous literature [12,22].

### 3.2. Sealed Gas Effect Analysis

As simulated in reference [12], the applied pressure difference will induce a deflection of the graphene membrane at the hundred-nanometer scale. If the depth of the pressure cavity is only several hundred nanometers, the volume change of the sealed gas in the cavity is no longer negligible. To the best of our knowledge, this problem has not been carefully analyzed. Herein, we tested the PMMA/Gra pressure sensors with the 285-nm-deep cavity for comparison. Figure 7 shows the time response of sensor resistance under different pressure differences (black curve). Even though the sensor response still follows the pressure change (red curve), its stability and repeatability are much worse than that of the through-hole sensor (in Figure 5). Further, we extracted its sensitivity S = $5.99\times 10^{-5}$/kPa, which is about 24% lower than the through-hole sensor. 

We attribute the sensitivity difference to the volume change of the sealed gas in the 285-nm-deep pressure cavity. By taking the deflection-induced volume change into consideration, an improved piezoresistive model is derived in this paper. 

First, the gauge factor (GF), a measure of piezoresistive effect, is defined as the ratio of relative change in electrical resistance *R* to the strain ε:(2)$$GF=\frac{∆R/R}{\varepsilon },$$

According to the theoretical analysis and experiments in reference [22], the GF is strain-independent and is in the range from 1.25 to 7 for the graphene membrane. In this paper, we choose the lower limit of GF=1.25 conservatively. 

The relation of the pressure *P* and deflection *d* for a circular membrane is given as [27]: (3)$$\left|P-P_{ref}\right|=\frac{8Etd^{3}}{3\left(1-\upsilon \right)r^{4}}+\frac{4td}{r^{2}}\sigma_{0},$$
where $P_{ref}$ is the reference pressure sealed in the cavities, *t* and *r* are the graphene thickness and radius, respectively, *E* and υ are Young’s modulus and Poisson’s ratio of graphene, respectively, $\sigma_{0}$ is the initial stress of the graphene membrane, and *d* is the deflection in the center of the membrane. Since the deflection is much smaller than the diameter of the suspended graphene membrane, an approximative relation of the strain and the deflection can be obtained [28].
(4)$$\varepsilon \approx \frac{2d^{2}}{3r^{2}},$$

By plugging Equations (2) and (4) into (3) to substitute the parameter *d*, we get the classical piezoresistive model as: (5)$$P=\{\begin{array}{c} P_{ref}+C_{1}{\left(\frac{∆R}{R}\right)}^{3/2}+C_{2}{\left(\frac{∆R}{R}\right)}^{1/2},\;P>P_{ref}\\ P_{ref}-C_{1}{\left(\frac{∆R}{R}\right)}^{3/2}-C_{2}{\left(\frac{∆R}{R}\right)}^{1/2},\;P<P_{ref} \end{array},$$
where
(6)$$C_{1}=\frac{4Et}{\left(1-\upsilon \right)r\cdot GF}\sqrt{\frac{3}{2GF}},$$
(7)$$C_{2}=\frac{4t\sigma_{0}}{r}\sqrt{\frac{3}{2GF}},$$

In most literature, the $P_{ref}$ in Equation (4) is considered as constant. This is valid for the through-hole pressure sensor, but failed in sensors with the finite-depth cavity. The reality is that the reference pressure is $P_{ref}=P_{0}+\Delta P_{0}$, where $P_{0}=1\;\mathrm{atm}$, and ΔP is the change of reference pressure due to the volume change ΔV under a membrane deflection *d*. Herein, we accurately calculate the volume change ΔV [29] in cavities when the pressure is in equilibrium on both sides of the graphene membrane.
(8)$$\Delta V=\frac{1}{6}\pi d\left(3r^{2}+d^{2}\right)$$

According to the equation of state of a hypothetical ideal gas (Boyle’s law) [30], the pressure change induced by the volume change is derived as:(9)$$P_{0}V_{0}=\left(P_{0}+\Delta P_{0}\right)\left(V_{0}+\Delta V\right)$$
where $V_{0}=\pi r^{2}h$. By plugging (2), (4) and (8) into it and simplifying, we obtain $\Delta P_{0}$ as:(10)$$\Delta P_{0}=\{\begin{array}{c} \frac{-P_{0}}{1-\frac{4h}{d\left(2+\frac{∆R/R}{GF}\right)}},\;P>P_{ref}\;\\ \frac{-P_{0}}{1+\frac{4h}{d\left(2+\frac{∆R/R}{GF}\right)}},\;P<P_{ref} \end{array}$$
where $d=r\sqrt{\frac{3}{2GF}}{\left(\frac{∆R}{R}\right)}^{1/2}$. Substituting $P_{ref}$ into Equation (4), the improved piezoresistive model is
(11)$$P=\{\begin{array}{c} P_{0}+\Delta P_{0}+C_{1}{\left(\frac{∆R}{R}\right)}^{3/2}+C_{2}{\left(\frac{∆R}{R}\right)}^{1/2},\;P>P_{ref}\\ P_{0}+\Delta P_{0}-C_{1}{\left(\frac{∆R}{R}\right)}^{3/2}-C_{2}{\left(\frac{∆R}{R}\right)}^{1/2},\;P<P_{ref} \end{array},$$

The relative resistance change ratio ∆R/R was calculated under different pressure differences for pressure sensors with several cavity depths, as shown in Figure 8a. Further, the corresponding sensitivity defined in Equation (1) was extracted in Figure 8b. It is clear that the cavity depth has a considerable effect on the performance of the pressure sensor. In general, for both the electrical response and the sensitivity, the smaller the depth is, the worse the performance becomes. When the cavity is close to the through-hole, the pressure sensor will achieve a symmetric response for both the bulging side and compressive side, and exhibits almost the same behavior as the classical model, which indicates that, for the deep-hole or through-hole pressure sensors, the sealed gas effect can be negligible. As the cavity becomes shallow, the sealed gas in the cavity plays three major influences. First, the electrical signal ∆R/R becomes weaker, which brings the difficulty in the electrical measurements. Additionally, it severely deteriorates the sensitivity at the low-pressure regime. Therefore, in Figure 8b, the corresponding sensitivity curves show deep valleys near zero pressure difference. Thirdly, the sealed gas affects the compressive side much more than the bulging side, which results in the asymmetry in both the resistance response and the corresponding sensitivity curve. At the compressive side, the sensitivity for the sensor with a 285-nm-deep cavity is at least 10 times smaller than that of the sensor with a through-hole. The proposed new piezoresistive model exhibits a deeper insight into the pressure sensing mechanism. It is a helpful tool for better performance pressure sensor design and characteristic analysis. 

## 4. Conclusions

The high-yield fabrication process by utilizing PMMA to support and seal the atomic graphene sheet was applied to two different pressure sensor structures: through-hole type and finite-depth type. The PMMA/graphene pressure sensor with the through-hole cavity operates as fast as the commercial pressure sensor. Even though the PMMA/graphene membrane is hundreds of times thicker than single layer graphene, our sensor exhibits a comparable sensitivity per unit area to the state of the art of the graphene pressure sensor without the PMMA supporting layer, which hints that the gas leakage from cracks or defects of the graphene sheet severely affects the device performance. Further, the performance of PMMA/graphene pressure sensors with a 285-nm-deep cavities and through-hole cavities are comparatively studied. The device with the 285-nm-deep cavity shows worse sensitivity and stability due to the volume-induced pressure change of the sealed gas in the pressure cavity. By taking the pressure change into consideration, an improved piezoresistive model is derived. The results deduced from the new model are consist with our experimental results. Our findings point out a promising route for better performance of graphene-based pressure sensors.

## Figures and Tables

**Figure 1 micromachines-11-00786-f001:**
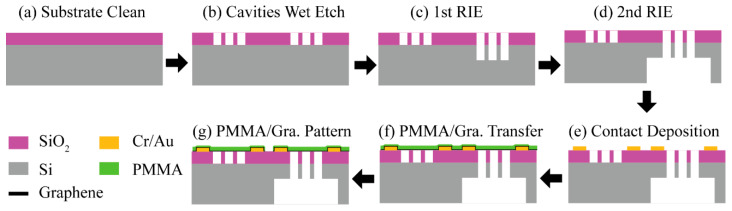
Schematic illustration of the fabrication process.

**Figure 2 micromachines-11-00786-f002:**
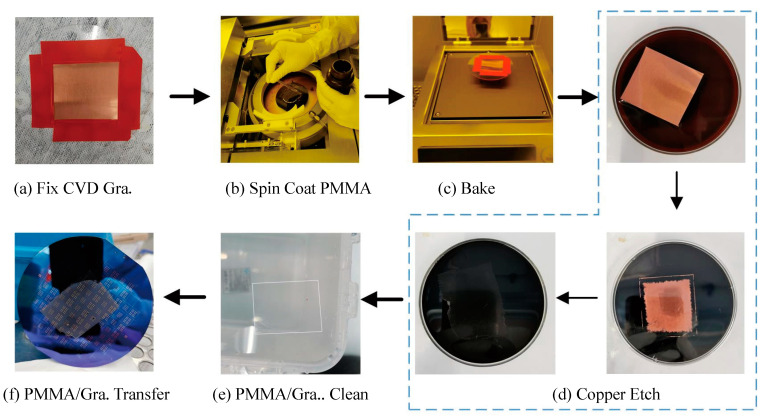
Polymethylmethacrylate (PMMA)/graphene heterostructure transfer flow.

**Figure 3 micromachines-11-00786-f003:**
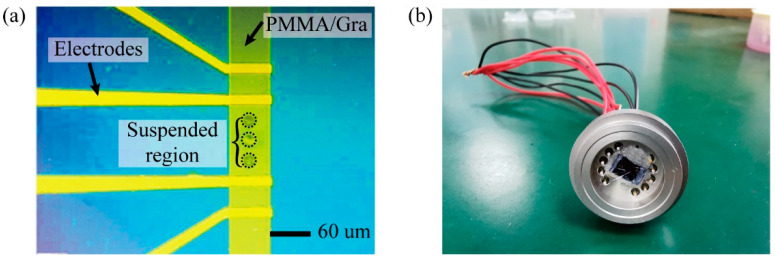
(**a**) Optical image of fabricated device; (**b**) Packaged device for pressure sensing.

**Figure 4 micromachines-11-00786-f004:**
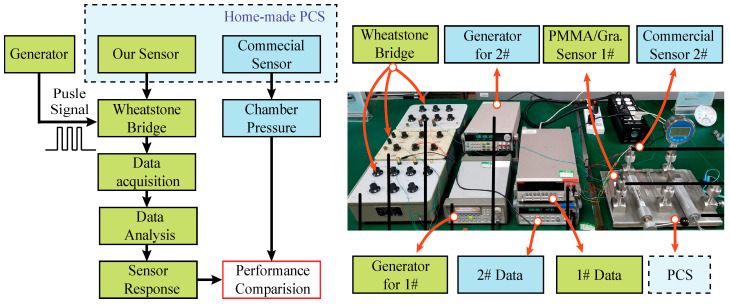
Measurement scheme and corresponding homemade setup.

**Figure 5 micromachines-11-00786-f005:**
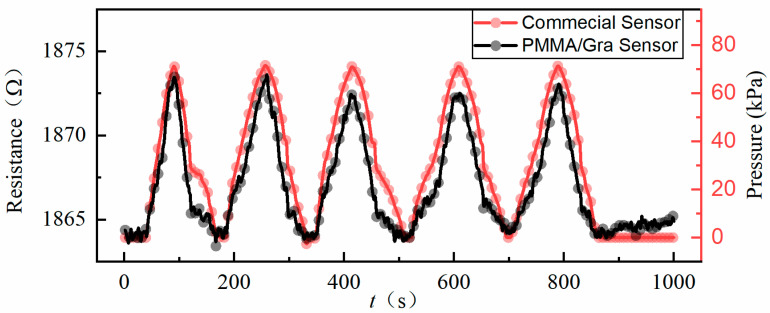
Measured resistance versus time during pressure cycles: the dark line corresponds to the PMMA/graphene pressure sensor with through-hole, the red line is the commercial sensor.

**Figure 6 micromachines-11-00786-f006:**
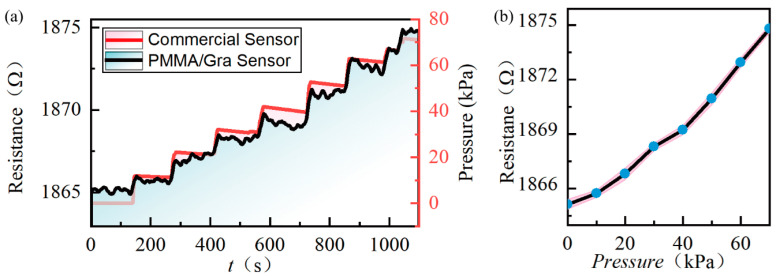
(**a**) Sensor response vs. time under difference pressure steps; (**b**) Sensor resistance vs. applied pressure difference—the light red envelope represents the deviation of sensing fluctuation.

**Figure 7 micromachines-11-00786-f007:**
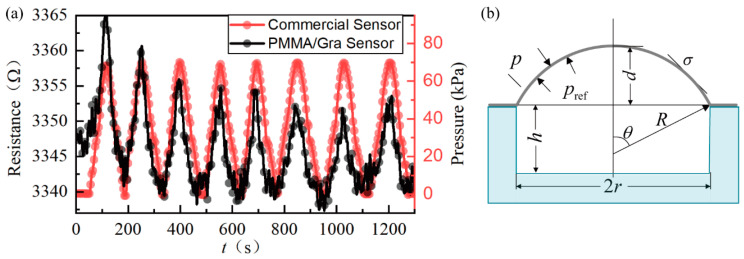
(**a**) The time-dependent responses of the PMMA/Gra pressure sensors with the 285-nm-deep cavity. (**b**) Schematic of the pressure sensor.

**Figure 8 micromachines-11-00786-f008:**
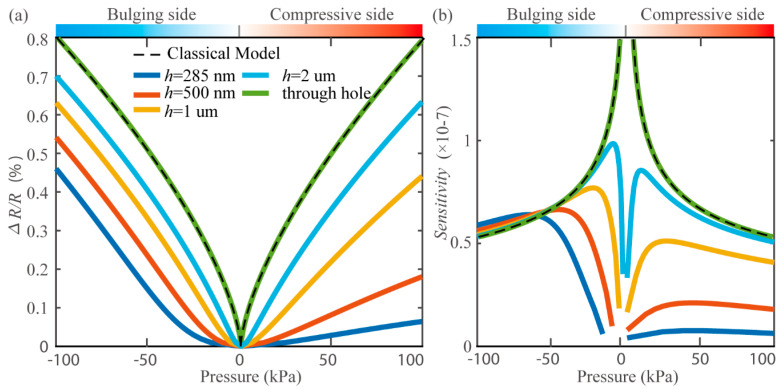
(**a**) Resistance change ratio Δ*R*/*R* as a function of applied pressure difference. (**b**) Corresponding sensitivity extracted from the curves in (**a**).

**Table 1 micromachines-11-00786-t001:** Summary of typical piezoresistive membrane-based pressure sensors.

Group	Year	Material	Size μm	Range kPa	Sensitivity /kPa	Sensitivity Per Unit Area/kPa
Melvas [23]	2002	Si	100 × 100	10~140	3.75 × 10^−5^	3.75 × 10^−9^
Hierold [24]	2007	Al_2_O_3_/CNT	*r* = 100	0~130	6.53 × 10^−5^	2.08 × 10^−9^
Gonzalez [25]	2012	SiGe	200 × 200	0~100	4.60 × 10^−5^	1.15 × 10^−9^
Godovitsyn [26]	2013	Si	2000 × 2000	0~100	2.37 × 10^−4^	5.92 × 10^−11^
Smith [12]	2013	Graphene	6 × 64	40~100	2.96 × 10^−5^	7.71 × 10^−8^
This paper	2019	Graphene	*r* = 8	0~70	7.42 × 10^−5^	12.30 × 10^−8^

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
