# Peer review of "Improved High-Yield PMMA/Graphene Pressure Sensor and Sealed Gas Effect Analysis"

_micromachines, 2020, doi:10.3390/mi11090786_

Round 1

Reviewer 1 Report

Dear Authors

I have a few remarks, comments, and suggestions on the manuscript:

  • Figure 2 is not directly addressed in the text. My proposal is, to insert sentences in Line 89: …… strength. The transfer flow of PMMA/Graphene heterostructure is presented in Figure 2. As shown ………..
  • Equations (5), (11), and (12): Please check. Maybe I am wrong, but in my opinion, P<Pref and P>Pref should be interchanged.
  • There is no label for Equation (10).

Most other remarks you can find as comments in the attached file (micromachines-892678-peer-review-v1_CORRECTION.pdf). English is not my first language so I am not qualified to correct it with authority. However, I tried to make some corrections in the attached PDF file. From this point of view, I recommend that the manuscript be corrected and linguistically improved by a native English speaker.

Reviewer 2 Report

Throughout, the use of articles such as the, a, an, etc. is not always standard. Some verbs are incorrectly used with the correct tense or agreement with the singular or plural subject. Please have a native English reader correct the grammar.

The optical image in Fig. 3a would benefit from labels identifying the sensor parts, as well as a scale bar for dimensions. A diagram of the sensor showing electrical components would be very useful. It appears that there are several features in the graphene region between the central contacts. What are these and what are their dimensions relative to the width of the graphene strip that makes up the sensor? How do these dimensions of the sensor piezoelectric region affect the accuracy of the calculations given later?

Line 36: “micros.” should be “microns.”

Line 121: “a clear nitrogen source...” does clear mean high-purity, regulated, or filtered?

Line 130: “output voltage is right the response” is unclear. What is meant?

Line 171: “The improved performance due to the leak-free implies that the leakage of graphene membrane may has much worst effect than what has been reported…” should be “The improved performance due to the absence of leaks implies that the leakage of graphene membrane may have much worse effect than what has been reported…”

Line 180: “the volume change of the sealed gas in the cavity is no longer nonnegligible.” Should this be “no longer negligible”?

Line 181: Rewrite as  “As far as we know, this problem has not been carefully analyzed.”

Round 2

Reviewer 2 Report

This revised manuscript meets the high standards of the journal and deserves publication. A few minor changes could be corrected in proof, as listed below. The Authors have made much improvement in English, but to an English speaker, some usage sounds strange and this can be easily corrected by following simple rules for readability. Some examples are listed here.

Plural nouns are preferred for describing a class of things, such as the objects of a study.

16: In this paper, we utilized PMMA/graphene heterostructures to isolate graphene from the ambient environment and to enhance its strength simultaneously. PMMA/graphene pressure sensors with finite-depth cavities and through-hole 18 cavities separately were made for comparative study.

253: point out a promising route for better performance of graphene-based pressure sensors.

The use of the article “the” is unnecessary before general terms of materials or indefinite quantities. The indefinite article is used to introduce a new topic. See http://www.butte.edu/departments/cas/tipsheets/grammar/articles.html

29: is an ideal option as a sensing membrane for pressure

73: etched by buffered oxide etchant (BOE)

86: the most important issues toward high yield

102: which consists of a LOR 10A 103 resist layer and a AZ 1500 resist layer

129: overwhelmed by thermal and electromagnetic noise.

151: relationship of the graphene resistance and applied pressure similar to most literature reports

The following sentence should point to improvement in sensitivity or other properties, as this would be more accurate than “better.”

164: Graphene-based pressure sensors are at least one order better than conventional silicon-based and CNT-based pressure sensors. Our PMMA/Gra pressure sensor is comparable and a little better than

Author Response

We thank the Referee for the help of the English improvement. We have accepted all the suggestions and modified the manuscript once again. The corrections are listed below:

Plural nouns are preferred for describing a class of things, such as the objects of a study.

16: In this paper, we utilized PMMA/graphene heterostructures to isolate graphene from the ambient environment and to enhance its strength simultaneously. PMMA/graphene pressure sensors with finite-depth cavities and through-hole 18 cavities separately were made for comparative study.

Response 1: We have corrected it in the manuscript line 18 and 19. (in red)

253: point out a promising route for better performance of graphene-based pressure sensors.

Response 2: We have corrected it in the manuscript line 256. (in red)

The use of the article “the” is unnecessary before general terms of materials or indefinite quantities. The indefinite article is used to introduce a new topic. See http://www.butte.edu/departments/cas/tipsheets/grammar/articles.html

29: is an ideal option as a sensing membrane for pressure

Response 3: We have corrected it in the manuscript line 30. (in red)

73: etched by buffered oxide etchant (BOE)

Response 4: We have corrected it in the manuscript line 74. (in red)

86: the most important issues toward high yield

Response 5: We have corrected it in the manuscript line 87. (in red)

102: which consists of a LOR 10A 103 resist layer and a AZ 1500 resist layer

Response 6: We have corrected it in the manuscript line 103 and 104. (in red)

129: overwhelmed by thermal and electromagnetic noise.

Response 7: We have corrected it in the manuscript line 130 and 131. (in red)

151: relationship of the graphene resistance and applied pressure similar to most literature reports

Response 8: We have corrected it in the manuscript line 152. (in red)

The following sentence should point to improvement in sensitivity or other properties, as this would be more accurate than “better.”

164: Graphene-based pressure sensors are at least one order better than conventional silicon-based and CNT-based pressure sensors. Our PMMA/Gra pressure sensor is comparable and a little better than

Response 9: It is the improvement of sensitivity. We have corrected it in the manuscript line 167. (in red)

Response 10: Except those corrections above, there are some similar corrections which are marked in red also.